# OpenReview forum: "Dataset Featurization: Uncovering Natural Language Features through Unsupervised Data Reconstruction"
_ICLR.cc/2026/Conference — ICLR 2026 Conference Desk Rejected Submission_

### Official Review · Reviewer_eKkX · 2025-10-25

**Soundness:** 3
**Presentation:** 2
**Contribution:** 2
**Rating:** 4
**Confidence:** 3

**Summary:**

The authors propose a domain-agnostic method to featurize text datasets with natural-language features. By leveraging a language model to produce possible features, and perplexity minimization during original text reconstruction to select them, the authors are able to extract interpretable features without supervision. The authors validate the proposed method on several dataset modeling tasks and case studies, demonstrating higher effectiveness in feature extraction compared to previous approaches.

**Strengths:**

- The authors test the method on several dataset-modeling tasks and case studies, validating it from multiple perspectives.
- The method is practical and valuable, yielding better performance than other baselines.

**Weaknesses:**

- It is unclear which failure modes arise in the generation stage and how they could be mitigated.
- Some claims and methodological choices are not clearly explained or insufficiently supported; I outline specific concerns in the questions section.

**Questions:**

- One concern about the method is that the generation stage may produce non-discriminative or overly generic features. Although the method is designed to discard irrelevant features because they would lead to high perplexity during text reconstruction, there can be features that actually maintain low perplexity across multiple samples while providing minimal information about the dataset's structure. In Section 6.2, the authors mention filtering low-variance features, suggesting this problem was encountered in practice, though this appears as a post-hoc fix rather than a solution. I would like to ask the authors whether they have encountered this type of issue systematically and whether it is possible to detect or prevent such features during the generation stage.
- In Section 6.2, the authors report that on HH-RLHF the method achieves 68.1% accuracy while the baseline reaches 68.9% (Table 2). In Appendix F.2, the authors attribute this to an accuracy ceiling due to dataset noise (potentially incorrect annotations). However, if the dataset noise is the limiting factor, why does the baseline with hand-crafted features perform slightly better? Does this suggest that the unsupervised features are more susceptible to noise?
- In Section 4, the authors mention setting C=5 and K=5 for all experiments and observing “additional performance gains by increasing the number of proposed features,” and refer to Appendix D.1 for additional details.  I couldn't find an ablation study systematically varying C and K. I would like to ask the authors whether they explored how performance scales with these hyperparameters?
- Considering the greedy sequential selection in the featurization stage: since the choice of earlier features affects which subsequent features provide marginal information gain, does the order of evaluation significantly impact which features are selected? Does running the pipeline multiple times produce consistent feature sets?

---

> ### Author Response · Authors · 2025-11-21
> **Author's Response - Thank you for your feedback (1/3)**
>
> We thank the reviewer for his review and we appreciate that they have found our method **practical and valuable**, and **praised our evaluation framework** consisting of multiple dataset-modeling tasks. We now address the reviewer’s specific concerns and questions.
>
> > _**One concern about the method is that the generation stage may produce non-discriminative or overly generic features. … In Section 6.2, the authors mention filtering low-variance features, suggesting this problem was encountered in practice, though this appears as a post-hoc fix rather than a solution. I would like to ask the authors whether they have encountered this type of issue systematically and whether it is possible to detect or prevent such features during the generation stage.**_
>
> We appreciate the opportunity to clarify the distinction between "generic" and "low-variance" features within the specific context of Preference Modeling. In our standard pipeline, broad features, such as "The text is a review," **do not represent a failure mode**. Instead, the greedy reconstruction objective naturally selects these features early to establish a "centroid" for the data distribution before shifting toward more discriminative features that reduce residual perplexity. This behavior could potentially be prevented by initializing the model within this centroid from the start, for example, by prompting it with a generic dataset description. However, **our evaluation framework demonstrates that this step is not necessary** to outperform the baselines.
>
> Regarding the specific filtration mentioned in Section 6.2, it was not applied as a post-hoc correction during the generation stage but **rather was related to modality conversion**. In that experiment, binary features were converted into continuous ratings on a 0–10 scale to align with the baseline methodology. Features that might be informative in a binary context (e.g., "The response contains only one sentence") can lose their informative power when projected onto a continuous scale. Therefore, we removed these features (less than 10% of the total features) solely to ensure the regression model received meaningful variance. We **did not encounter this issue** in our other case study or the dataset modelling experiments.
>
> We will clarify in the manuscript that this filtration specifically addresses the continuous rating conversion in the Preference Modeling application and does not represent a limitation inherent to the core binary pipeline.
>
> > _**In Section 6.2, the authors report that on HH-RLHF the method achieves 68.1% accuracy while the baseline reaches 68.9% (Table 2). In Appendix F.2, the authors attribute this to an accuracy ceiling due to dataset noise (potentially incorrect annotations). However, if the dataset noise is the limiting factor, why does the baseline with hand-crafted features perform slightly better? Does this suggest that the unsupervised features are more susceptible to noise?**_
>
> We believe the slight underperformance compared to expert features on HH-RLHF (68.1% vs 68.9%) stems from **a fundamental difference in optimization objectives**. Our unsupervised method optimizes for reconstruction, meaning it aims to **faithfully fit the entire data distribution**, including its inherent noise and annotation artifacts. In contrast, expert features are manually engineered to align with "correct" preferences, implicitly ignoring data noise that does not contribute to the desired signal. Consequently, our method effectively captures the dataset's true distribution, including the noise in HH-RLHF label, which results in a slight penalty on the accuracy metric compared to features designed to ignore that noise. However, we emphasize that matching human performance within 1% without any manual engineering demonstrates the **method's high utility**. We will revise Section 6.2 to explicitly discuss this trade-off between reconstruction fidelity and noise robustness.

---

> ### Author Response · Authors · 2025-11-21
> **Author's Response - Thank you for your feedback (2/3)**
>
> > _**In Section 4, the authors mention setting C=5 and K=5 for all experiments and observing “additional performance gains by increasing the number of proposed features,” and refer to Appendix D.1 for additional details. I couldn't find an ablation study systematically varying C and K. I would like to ask the authors whether they explored how performance scales with these hyperparameters?**_
>
> We apologize for the omission of these specific results in the main text. To address this, **we conducted an ablation study on the Amazon dataset** using the Qwen 0.5B (base) model to evaluate the impact of varying the number of generated features per comparison ($K$) and the number of context examples ($C$).
>
> We evaluated performance with $K \in \{5, 7, 10\}$ while keeping $C=5$. As shown in the Table below, **increasing $K$ yields consistent improvements across all metrics**. A larger $K$ provides the selection stage with a richer pool of candidate features, allowing the greedy optimization to find better fits. However, we utilized $K=5$ in the main paper as it offers **a strong balance between performance and generation costs**; while $K=10$ performs best, $K=5$ achieves competitive results with half the token consumption during the proposal phase.
>
> | $K$ (Feats) | Class Coverage | | | Recon. Accuracy | | | Semantic Pres. | | |
> | :--- | :---: | :---: | :---: | :---: | :---: | :---: | :---: | :---: | :---: |
> | | Top 10 | Top 20 | Top 50 | Top 10 | Top 20 | Top 50 | Top 10 | Top 20 | Top 50 |
> | **5** | 0.524 | 0.578 | 0.593 | 0.646 | 0.710 | 0.742 | 1.56 | 2.33 | **2.78** |
> | **7** | 0.528 | 0.578 | 0.604 | 0.646 | 0.715 | 0.754 | 1.67 | 2.33 | 2.33 |
> | **10** | **0.586** | **0.613** | **0.618** | **0.681** | **0.729** | **0.757** | **2.00** | **2.67** | 2.67 |
>
> We also evaluated performance with $C \in \{5, 7, 10\}$ while keeping $K=5$. As shown in the table below, **increasing the number of comparison texts did not yield linear improvements**. We hypothesize this is because our current evaluation metrics target high-level class reconstruction (e.g., "Is this a Book review?"). While providing more context ($C=10$) likely leads to more nuanced or granular feature proposals, **these subtleties may not result in higher scores on broad category-based metrics** compared to the high-level features already found at $C=5$.
>
> | $C$ (Exs) | Class Coverage | | | Recon. Accuracy | | | Semantic Pres. | | |
> | :--- | :---: | :---: | :---: | :---: | :---: | :---: | :---: | :---: | :---: |
> | | Top 10 | Top 20 | Top 50 | Top 10 | Top 20 | Top 50 | Top 10 | Top 20 | Top 50 |
> | **5** | **0.524** | 0.578 | 0.593 | **0.646** | 0.710 | **0.742** | 1.56 | **2.33** | **2.78** |
> | **7** | 0.480 | 0.539 | 0.545 | 0.618 | 0.699 | 0.717 | **1.67** | 2.00 | 2.67 |
> | **10** | 0.521 | **0.579** | **0.597** | 0.633 | **0.726** | 0.733 | 1.33 | 2.00 | 2.33 |
>
> We thank the reviewer for highlighting this necessary analysis and will include these tables and the associated discussion in Appendix D.1.

---

> ### Author Response · Authors · 2025-11-21
> **Author's Response - Thank you for your feedback (3/3)**
>
> > _**Considering the greedy sequential selection in the featurization stage: since the choice of earlier features affects which subsequent features provide marginal information gain, does the order of evaluation significantly impact which features are selected? Does running the pipeline multiple times produce consistent feature sets?**_
>
> Regarding the stability of the pipeline, it is important to **distinguish between the two stages**. While the Generation Stage is stochastic (the initial pool of proposed features varies based on the LLM's sampling), the Selection Stage is deterministic: given a fixed set of candidates, our greedy algorithm always selects the same feature set. To empirically test end-to-end consistency, we executed the full pipeline **three independent times on the Amazon dataset**. We performed this evaluation across varying model sizes (Qwen 0.5B, 3B, and 7B) to assess how **model capability influences stability**.
>
> As shown in the table below, smaller models exhibit higher variance, which we attribute to their weaker language modeling capabilities rendering them more susceptible to noise in the initial feature proposal stage. However, **this variance significantly decreases as model scale increases**. For the 7B model, the confidence intervals are tight, indicating high stability. This confirms that while the specific lexical form of selected features may vary across runs (e.g., one run selecting "Topic is Sports" and another "Discusses athletic events"), **the semantic subspace they span and the resulting performance metrics remain highly consistent**. We appreciate this suggestion and will include it in Appendix D.1.
>
> | Model      | Class Coverage ↑ (mean ± CI) |                             |                             | Reconstruction Accuracy ↑ (mean ± CI) |                             |                             | Semantic Preservation ↑ (mean ± CI) |                             |                             |
> |-----------|------------------------------:|----------------------------:|----------------------------:|--------------------------------------:|----------------------------:|----------------------------:|------------------------------------:|----------------------------:|----------------------------:|
> |           | Top 10                        | Top 20                      | Top 50                      | Top 10                               | Top 20                      | Top 50                      | Top 10                             | Top 20                      | Top 50                      |
> | Qwen 0.5B | 0.524 ± 0.080                 | 0.578 ± 0.044               | 0.593 ± 0.046               | 0.646 ± 0.080                         | 0.710 ± 0.071               | 0.742 ± 0.069               | 1.555 ± 0.957                       | 2.333 ± 0.829               | 2.778 ± 0.957               |
> | Qwen 3B   | 0.543 ± 0.038                 | 0.571 ± 0.064               | 0.605 ± 0.007               | 0.668 ± 0.043                         | 0.714 ± 0.079               | 0.751 ± 0.016               | 1.778 ± 0.478                       | 2.444 ± 0.479               | 3.000 ± 0.000               |
> | Qwen 7B   | 0.558 ± 0.026                 | 0.583 ± 0.051               | 0.601 ± 0.023               | 0.654 ± 0.027                         | 0.723 ± 0.034               | 0.756 ± 0.011               | 2.000 ± 0.478                       | 2.444 ± 0.000               | 2.444 ± 0.000               |
>
> &nbsp;
>
> We hope these clarifications regarding feature specificity, noise robustness, and stability address your concerns and encourage you to reconsider your evaluation towards acceptance.

---

### Official Review · Reviewer_MrYc · 2025-10-31

**Soundness:** 3
**Presentation:** 3
**Contribution:** 3
**Rating:** 6
**Confidence:** 3

**Summary:**

The paper proposes Dataset Featurization: a three-stage pipeline that (1) elicits many human-readable binary features via an LLM, (2) de-duplicates them with embedding-based clustering and obtains per-example truth values, and (3) greedily builds a global feature set that minimizes the mean per-text perplexity of a scoring LM when conditioned on the features that are true for each example. The resulting small, interpretable “feature basis” is shown useful in two applications: (i) jailbreak tactic compression (achieving comparable effectiveness to a 500-cluster baseline with ~20 features across multiple target models), and (ii) composable preference modeling (CPM) on SHP/HH-RLHF, where automatically discovered features rival or surpass expert-crafted ones at similar feature counts.

**Strengths:**

* Clear, modular pipeline with a principled dataset-level objective and an interpretable feature representation.

* Demonstrates strong compression of jailbreak strategies while maintaining attack efficacy across multiple target models.

* Shows that automatically discovered features can match/beat expert-designed features in CPM, with reasonable robustness analyses.

* Practical engineering details.

**Weaknesses:**

* Binary-only features and reliance on positive instances during optimization limit expressivity.

* The optimization is global, so it does not guarantee per-instance optimality; some examples may not benefit from the selected global basis.

* No dedicated length-sensitivity/long-context study (data filtered to a range; but scalability to very long inputs remains unclear).

**Questions:**

* What is the length sensitivity of the scoring objective e.g. does performance degrade near the upper character limit or with longer contexts?

* Will you release prompts, code, and feature lists to facilitate replication?

---

> ### Author Response · Authors · 2025-11-21
> **Author's Response - Thank you for your feedback**
>
> We thank the reviewer for their feedback and appreciate that they found our pipeline implementation **clear,** our analyses **reasonable**, and valued the inclusion of **practical engineering details**. We address your specific questions below.
>
> > _**Binary-only features and reliance on positive instances during optimization limit expressivity.**_
>
> We acknowledge that binary features are a constraint compared to continuous representations. However, our Preference Modeling case study demonstrates that these binary features often serve as robust proxies for continuous signals. In that experiment, we **successfully converted our binary features into 0-10 ratings by simply instructing an LLM to produce a scalar** instead of a binary value, matching the performance of human-crafted features. While we filtered a small fraction (~10%) of features that lacked variance on a continuous scale to achieve a fully robust feature set, the **vast majority captured sufficient granular signal** to train a competitive reward model.
>
> Furthermore, our method could naturally be extended to handle multi-class or scalar-valued features. For example, instead of indicating only the presence or absence of a feature in the context prompt for dataset modeling, one could include a sentence specifying the feature’s value (whether a class label or a scalar). Nonetheless, we believe the current formulation offers a practical balance between **computational efficiency and expressivity for most downstream applications**, and leave these extensions for future work. We will clarify this trade-off in the limitations section.
>
> > _**The optimization is global, so it does not guarantee per-instance optimality; some examples may not benefit from the selected global basis.**_
>
> We agree that a global feature set effectively acts as a dimensionality reduction technique, which inherently involves some information loss at the instance level. However, we view this as a feature of the method rather than a bug: **our goal is to discover a unified interpretable basis** that describes the dataset as a whole, rather than optimizing for individual samples in isolation. This ensures that the resulting features allow for systematic comparison across the entire corpus, similar to how PCA identifies global principal components.
>
> > _**What is the length sensitivity of the scoring objective e.g. does performance degrade near the upper character limit or with longer contexts?**_
>
> We argue that our method is robust across varying text lengths. Our experimental design **deliberately included datasets with substantial length variability**, such as mixtures of shorter and longer New York Times articles (ranging from a few hundred characters up to 10,000; see table below). Because our method consistently outperformed the baselines across these diverse datasets, we did not observe any degradation related to text length within standard context windows.
>
> | Percentile (sample character length) | AMAZON length | NYT length | DBPEDIA length |
> |---------------------------:|--------------:|-----------:|----------------:|
> | 10th                       | 121           | 426        | 145             |
> | 25th                       | 161           | 1070       | 231             |
> | 50th                       | 256           | 3762       | 402             |
> | 75th                       | 462           | 5734       | 740             |
> | 90th                       | 870           | 7173       | 1257            |
>
> We hypothesize that the upper bound of performance is dictated not by our scoring objective, but by **the underlying LLM's ability to model long-context dependencies** (e.g., the "Lost in the Middle" phenomenon [1]). As such, our method scales with the capabilities of the backbone model. We will include the specific length statistics of our datasets in the appendix to clarify the range in which the method was validated.
>
>
> > _**Will you release prompts, code, and feature lists to facilitate replication?**_
>
> Yes. We have included all code and datasets in the supplementary materials provided with this submission. Upon acceptance, we are committed to releasing the full codebase, prompts, and extracted feature lists on public repositories (e.g., GitHub, HuggingFace) to ensure easy reproducibility.
>
> &nbsp;
>
> We hope **these clarifications address your concerns**, highlight the robustness of our approach, and provide a compelling reason to positively reconsider your assessment.
>
>
> [1] Lost in the Middle: How Language Models Use Long Contexts, TACL

---

> > ### Comment · Reviewer_MrYc · 2025-11-27
> >
> > Thank you for the responses. I will keep my score.

---

> > > ### Author Response · Authors · 2025-11-27
> > > **Author Response**
> > >
> > > Thank you for your time, constructive feedback, and for helping us improve the paper!

---

### Official Review · Reviewer_a49K · 2025-10-31

**Soundness:** 3
**Presentation:** 3
**Contribution:** 3
**Rating:** 6
**Confidence:** 4

**Summary:**

The authors introduce a method for breaking datasets down into a granularly controllable number of features. The method works by first having an LLM propose features for each sample in an unlabelled dataset, then they deduplicate the set of features by clustering similar features together. They then evaluate the accuracy of each features by whether the features are useful to an LLM in reconstructing the original data samples labelled by that feature (measured by minimising perplexity)

The authors evaluate on three synthetic datasets (DBPedia, NYT, Amazon
Reviews) with known ground-truth labels, demonstrating superior performance over LLM prompting
baselines and Zhong et al. (2024). They also show the effectiveness of their method with two case studies. For the first, they compress 500 jailbreak attack tactics into 20 features while maintaining attack
effectiveness and diversity. For the second case study, they automate preference model feature discovery, matching human-crafted features from Go et al. (2024) in accuracy and robustness.

**Strengths:**

The authors' perplexity / reconstruction-based optimisation framing seems novel and their results suggest that it's effective. Their case studies show real applications that are highly relevant to LLM safety and alignment. Extracting compact representations of jailbreaks from a large unlabelled set of attacks without supervision seems highly useful for understanding new attack strategies at scale. The case study on preference modeling also emphasises that the lack of supervision needed for this method might also mitigate bias arising from human labelling, but I'm less sold that the bias reduction point is the most groundbreaking feature of the method (not that the authors imply this). I'm not totally sure how to estimate the impact of this method - it genuinely seems useful across a wide set of tasks and domains, and the controllability and unsupervised nature of the method make it preferable over other methods. The case studies are helpful for understanding the sorts of tasks on which the authors expect the method to be impactful.

The experiments are well substantiated. The synthetic datasets they use are diverse, and the two case studies further illustrate how widely-applicable their method is. They compare to reasonable baselines in prompting and Zhong et al 2024. It's a little hard to know how much further you could push the prompt engineering, but it intuitively seems like prompting would still be overall worse than a pipeline like Zhong et al or from this paper. They also convincingly demonstrate that their method improves over human-written features, which seems like another important axis of comparison for their method. The paper's extensive appendices leave very little unaddressed. I particularly appreciated the cost analysis in appendix C, which is a practical and thoughtful inclusion - though I would have appreciated just a little more depth in the cost comparison with Zhong et al, rather than just the (probably approximately true) claim that the grid search would increase the total cost and runtime of the competitor method tenfold.

Overall, the writing and presentation of the paper is clear, well motivated, and well organised.

**Weaknesses:**

Overall I think the paper is quite strong, so struggling to find substantive weaknesses.

It's worth calling out that the paper overall provides a useful advancement with this reconstruction focus on unsupervised dataset feature labelling, but as the authors themselves acknowledge, the paper is in large part building on other recent work (Zhong et al. 2024, Findeis et al. 2024, Go et al. 2024). In their experiments comparing their method to Zhong et al, the authors convinced me that their method is stronger and likely more efficient (in dollars and time), but the overall place it seems like this paper sits in the literature is as a noticeable but not paradigm-shifting improvement on a reasonably important and wide-ranging problem. So, while the paper is still original and significant, it's not superlative on either axis.

As the authors themselves mention in section 7, this pipeline is restricted to binary features, and it's probably true that a method that could extract richer feature representations while holding all else equal would be useful across a wider range of domains & tasks.

The authors also note in section 8 that their greedy method introduces potential convergence to local optima, so more investigation into alternative selection strategies could plausibly improve on this method? Would be interested in the authors' takes.

**Questions:**

1.) I'm pretty confused about how to think about the novelty/originality of the paper - could the authors make a.) the concise steelman for why this is work is highly original, or just b.) what their all-things-considered view is, here.
2.) How important is the limitation of this method only working for binary features? Just how many applications would open up?
3.) What did you do that convinced you that the greedy optimisation target was ~fine, and unlikely to get stuck in local optima?

---

> ### Author Response · Authors · 2025-11-21
> **Author's Response - Thank you for your feedback**
>
> We thank the reviewer for their thoughtful and encouraging review, particularly for assessing the paper as **quite strong,** the writing as **clear,** and the framing as **novel.**
>
> > _**Could the authors make a.) the concise steelman for why this is work is highly original, or just b.) what their all-things-considered view is, here.**_
>
> Our view is as follows: Prior approaches (like Zhong et al., 2024) treat feature extraction as a clustering problem, forcing data into non-overlapping buckets based on embedding distance. Our method **redefines feature extraction as a language modeling problem**. By optimizing for reconstruction (perplexity) rather than cluster separation, we remove the structural priors that limit other methods. This shift allows our pipeline to discovery overlapping and multi-granular features simultaneously (e.g., capturing both high-level topics and low-level stylistic nuances in the same pass), which is impossible for clustering-based methods. We believe this move from extrinsic constraints (clusters) to intrinsic data verification (reconstruction) is a **significant leap within this domain of unsupervised dataset modeling** and will inspire further progress in this area. Further, our verification of the **method’s efficacy through both dataset modelling experiments and two down-stream use-cases** shows that our approach works effectively in practice, justifying the significance of our contribution.
>
> > _**How important is the limitation of this method only working for binary features? Just how many applications would open up?**_
>
> We acknowledge that binary features are a constraint, but we view this as **a tractable trade-off rather than a barrier to utility**. As demonstrated in our Preference Modeling case study, binary features often carry sufficient signal to approximate continuous properties. In that experiment, **we converted binary outputs into 0–10 scalar ratings simply by instructing the LLM to produce a scalar** instead of a binary label for each feature. Although we subsequently filtered out a small portion (~10%) of features that showed insufficient variance on this scale, the vast majority provided enough granularity to match the performance of human-crafted features.
>
> Our method could also be **extended to multi-class or scalar-valued features**, a direction we leave for future work. For instance, rather than indicating the presence or absence of a feature in the context prompt for dataset modeling, one could include a sentence specifying the feature’s value (whether a class label or a scalar). However, across the use cases we examined, the binary abstraction consistently offered **the best balance of cost, interpretability, and expressiveness**. We will clarify this trade-off in the manuscript.
>
> > _**What did you do that convinced you that the greedy optimisation target was ~fine, and unlikely to get stuck in local optima?**_
>
> We initially shared the reviewer's concern regarding local optima. However, empirical evidence suggests the optimization landscape for this task is surprisingly smooth. Across our experiments, over **90% of runs successfully identified more than 50 features (averaging ~100)** without premature convergence. Even in the rare cases where the algorithm terminated early, it was only after selecting 40+ features, suggesting the method had already captured the bulk of the dataset's information. We interpret this to mean that while the greedy approach theoretically risks local optima, the **sheer number of candidate features generated by the LLM ensures there is almost always a "valid next step"** available to reduce perplexity. Given this reliable behavior, we opted to keep the algorithm greedy to **minimize computational overhead** rather than implementing more complex search strategies (like beam search) that yielded diminishing returns during development. However, we expect future work could improve our method by using more complex search strategies. We will highlight this in the manuscript.
>
> &nbsp;
>
> We hope these clarifications highlight the **robustness of our approach** and support a move toward a full acceptance.

---

### Official Review · Reviewer_efCi · 2025-10-31

**Soundness:** 1
**Presentation:** 1
**Contribution:** 2
**Rating:** 2
**Confidence:** 2

**Summary:**

This research tries to construct interpretable features for a textual corpus in an unsupervised manner. To do so, the researchers propose to first prompt LLMs to generate the features from the samples, and then confirm the effectiveness of the features by minimizing the perplexity of the dataset.

**Strengths:**

The high-level idea of using extracted features to reconstruct the dataset for selecting features is interesting.

**Weaknesses:**

1. The writing of this paper is terrible. It is pretty hard for me to get an understanding of how it works and why it works. For example, I have no idea how you compute the perplexity exactly from lines 170-178.  Are you following the contents in lines 140-145? If yes, how are these conditional PPLs measured concretely? Similarly, in Section 5, I don't know why we need to consider the class labels provided by the datasets, and how you evaluate your results. More importantly, why is this experiment result correct?

2. Experiments miss many simple but effective strategies to mine superficial features from the dataset unsupervisely. For example, you can compute the TF-IDF scores for different N-grams.

**Questions:**

Please see the weaknesses.

---

> ### Author Response · Authors · 2025-11-21
> **Author's Response - Thank you for your feedback**
>
> We thank the reviewer for their feedback. We believe there are several potential concerns regarding our experimental setup and metrics which we clarify below.
>
> >  _**The writing of this paper is terrible. It is pretty hard for me to get an understanding of how it works and why it works. For example, I have no idea how you compute the perplexity exactly from lines 170-178. Are you following the contents in lines 140-145? If yes, how are these conditional PPLs measured concretely?**_
>
> The reviewer asks how **conditional PPL** is measured concretely. As defined in **Section 3 (Dataset Modeling)** and explicitly detailed in **Algorithm 1**, we utilize standard conditional language modeling. For a given text $x$, we identify the set of features $\phi(x)$ that evaluate to true and concatenate their natural language descriptions (e.g., "The text is positive\nThe text mentions a book") to form a context prompt $C$. We then calculate the **perplexity** of the original text $x$ conditioned on this context $C$. Our optimization objective is simply to identify the feature set that **minimizes** this reconstruction perplexity. We will revise **Section 3** to make this mechanical step more explicit to prevent future confusion, and thank the review for their suggestions.
>
> > _**Similarly, in Section 5, I don't know why we need to consider the class labels provided by the datasets, and how you evaluate your results. More importantly, why is this experiment result correct?**_
>
> Our method is entirely **unsupervised** and never sees these labels during training/feature extraction. We use the dataset labels strictly as **Ground Truth** for validation (**Section 5.1**), following standard protocols for evaluating unsupervised learning (analogous to using labels to calculate Purity or NMI for clustering algorithms). The fact that our unsupervised features align with these human-annotated labels (as measured through our metrics) is precisely what proves the experiment is "correct", as it demonstrates that our method recovers meaningful, ground-truth structure without being told what to look for.
>
> > _**Experiments miss many simple but effective strategies to mine superficial features from the dataset unsupervisely. For example, you can compute the TF-IDF scores for different N-grams.**_
>
> We thank the reviewer for their suggestion. To address the reviewer's concern, we implemented a **TF-IDF baseline**. We used Scikit-learn’s TfidfVectorizer with Spacy tokenization (filtering stopwords/punctuation) to identify the top 50 unigrams and bigrams ranked by their mean TF-IDF score (min_df=5, max_df=0.9). To make these comparable to our method, we converted the extracted n-grams into natural language predicates (e.g., "The text mentions '[n-gram]'").
>
> | Method | Number of Features | **DBPEDIA** ||| **NYT** ||| **AMAZON** |||
> |---|---|---|---|---|---|---|---|---|---|---|
> |  |  | Class Coverage | Reconstruction Accuracy | Semantic Preservation | Class Coverage | Reconstruction Accuracy | Semantic Preservation | Class Coverage | Reconstruction Accuracy | Semantic Preservation |
> | **Dataset Featurization** | 5  | 0.771 | 0.913 | 2.000 | 0.461 | 0.599 | 0.333 | 0.470 | 0.583 | 2.000 |
> |  | 10 | 0.883 | 0.981 | 3.333 | 0.491 | 0.677 | 0.333 | 0.510 | 0.613 | 2.000 |
> |  | 20 | 0.896 | 0.991 | 3.667 | 0.696 | 0.849 | 0.667 | 0.613 | 0.728 | 2.333 |
> |  | **50** | **0.929** | **0.993** | **4.333** | **0.723** | **0.865** | **1.000** | **0.632** | **0.769** | **3.667** |
> | **TF-IDF** | 5  | 0.736 | 0.883 | 1.000 | 0.448 | 0.626 | 0.333 | 0.350 | 0.429 | 0.000 |
> |  | 10 | 0.825 | 0.975 | 1.000 | 0.501 | 0.688 | 0.667 | 0.382 | 0.497 | 0.000 |
> |  | 20 | 0.863 | 0.989 | 1.000 | 0.566 | 0.766 | 1.000 | 0.440 | 0.581 | 0.333 |
> |  | 50 | 0.891 | 0.990 | 1.667 | 0.609 | 0.833 | **1.000** | 0.513 | 0.675 | 0.667 |
>
> Our method matches or outperforms TF-IDF when using a small number of features, and it clearly surpasses TF-IDF once we scale to **50 features**, highlighting the strong scalability of our approach. We will include these results in the final manuscript.
>
> &nbsp;
>
> We believe these clarifications demonstrate the **soundness** of our methodology and the **validity** of our evaluation metrics. We hope these explanations resolve any concerns regarding the experimental setup and encourage you to reconsider your assessment.

---

> > ### Comment · Reviewer_efCi · 2025-11-27
> > **Official Comment to Author Response**
> >
> > Thanks for your clarification of my first concern, and I am OK with your response.
> >
> > However, I still hold concerns about the second Weakness I mentioned. With these additional results, we can observe that a simple baseline (i.e., TF-IDF) can practically extract meaningful lexical-level explanations for the classes. For example, on the DEPEDIA dataset, 50 features from TF-IDF can cover 89.1% of classes, while those from the proposed method can only improve to 92.9%. Similarly, the Reconstruction Accuracy between the proposed method and the baseline is only a 0.3% difference. These observation demonstrates that the tasks are too trivial for the evaluation since the features are just some superficial linguistic patterns that can even be captured by TF-IDF. The research can benefit from conducting additional experiments on more challenging datasets that require more semantic features for predictions.
> >
> > Overall, since the main claim of the research has been supported (although within this "toy" setting), I would raise my evaluation a little bit.

---

> ### Author Response · Authors · 2025-11-27
> **Author Response**
>
> We thank you again for your continued engagement and for updating your score. We agree that DBPEDIA is the simplest dataset in our suite from a lexical standpoint, but we believe it still provides important signal rather than constituting a trivial evaluation. While TF-IDF attains performance comparable to our method on some aggregate metrics, it does so primarily by exploiting surface n-grams. In contrast, our method achieves substantially higher Semantic Preservation on DBPEDIA (4.333 vs. 1.667 for TF-IDF), indicating that it captures class-level meaning in terms of human-interpretable concepts rather than just keyword overlaps.
>
> Moreover, DBPEDIA exposes a failure mode for more complex, clustering-based approaches. In our experiments, the Zhong et al. (2024) baseline underperforms specifically on DBPEDIA, with Class Coverage scores as low as 0.6 compared to >0.9 for our method. Thus, DBPEDIA is not “universally easy”: it is a setting where a simple lexical method (TF-IDF) happens to do well, but a sophisticated clustering-based method struggles, whereas our reconstruction-based approach remains robust. We therefore view DBPEDIA as a valuable “lexically strong” benchmark that complements the more semantically demanding Amazon and NYT datasets, on which the performance gaps between our method and TF-IDF widen further. We will clarify this role of DBPEDIA, and the contrasting behaviors of TF-IDF and Zhong et al., more explicitly in the final manuscript.

---

### Author Response · Authors · 2025-12-03
**Summary of Rebuttal for the AC**

During the rebuttal period, reviewers converged on five main technical concerns: (i) an unclear description of how conditional perplexity is computed and how labels are used (efCi); (ii) the absence of simple unsupervised baselines such as TF-IDF (efCi); (iii) worries about generic/low-variance features and whether this is a systematic failure mode (eKkX); (iv) a perceived limited originality gap versus Zhong et al. and related work (a49K); and (v) questions around greedy optimization, stability, and the hyperparameters C and K (a49K, eKkX). In response, we (1) clarified the conditional perplexity computation and the role of labels as only ground-truth for evaluation, not supervision, and will revise Sections 3/5 accordingly; (2) implemented a TF-IDF n-gram baseline, showing that our method matches or outperforms it with few features and clearly surpasses it at 50 features, especially on more semantic datasets (NYT, Amazon); (3) clarified that “low-variance feature” filtering occurs only when converting binary features to continuous scores in the preference-modeling case, not as a core pipeline fix; (4) added ablations over C and K showing performance improves with more proposed features (C) and is relatively stable with respect to the number of comparison examples (K); and (5) provided a stability analysis over multiple runs and model sizes, showing that while individual feature wordings change, the selected semantic subspace and metrics are consistent, especially for larger models.

We believe these clarifications and additional experiments address the main technical concerns and demonstrate that the method is sound, practical, and competitive or superior to baselines. Any remaining issues are primarily about scope (binary features, reconstruction-based objective) and dataset difficulty, rather than correctness.

---

### Note · Program_Chairs · 2026-01-17
**Submission Desk Rejected by Program Chairs**

The following references in this submission do not refer to real documents and/or have major errors in bibliographic information:

 B. Sorensen and Others. Clio: Privacy-preserving conversation analysis in large language models.